# Characterization of brain dystrophins absence and impact in dystrophin-deficient *Dmd*^mdx^ rat model

**Dorian Caudal**[ID][1]*, **Virginie François**[2], **Aude Lafoux**[1], **Mireille Ledevin**[3], **Ignacio Anegon**[4], **Caroline Le Guiner**[2], **Thibaut Larcher**[3], **Corinne Huchet**[1,2]

**1** Therassay Platform, CAPACITES, Université de Nantes, Nantes, France, **2** Nantes Gene Therapy Laboratory, Université de Nantes, INSERM UMR 1089, Nantes, France, **3** ONIRIS, INRA UMR 703, Nantes, France, **4** TRIP, INSERM UMR 1064-CRTI, Nantes, France

* dorian.caudal@capacites.fr

**Data Availability Statement:** All relevant data are within the manuscript and its Supporting Information files.

**Funding:** The author(s) received no specific funding for this work.

## Abstract

Duchenne Muscular Dystrophy (DMD) is a severe muscle-wasting disease caused by mutations in the DMD gene encoding dystrophin, expressed mainly in muscles but also in other tissues like retina and brain. Non-progressing cognitive dysfunction occurs in 20 to 50% of DMD patients. Furthermore, loss of expression of the Dp427 dystrophin isoform in the brain of *mdx* mice, the most used animal model of DMD, leads to behavioral deficits thought to be linked to insufficiencies in synaptogenesis and channel clustering at synapses. *Mdx* mice where the locomotor phenotype is mild also display a high and maladaptive response to stress. Recently, we generated *Dmd*^mdx^ rats carrying an out-of frame mutation in exon 23 of the DMD gene and exhibiting a skeletal and cardiac muscle phenotype similar to DMD patients. In order to evaluate the impact of dystrophin loss on behavior, we explored locomotion parameters as well as anhedonia, anxiety and response to stress, in *Dmd*^mdx^ rats aged from 1.5 to 7 months, in comparison to wild-type (WT) littermates. Pattern of dystrophin expression in the brain of WT and *Dmd*^mdx^ rats was characterized by western-blot analyses and immunohistochemistry. We showed that dystrophin-deficient *Dmd*^mdx^ rats displayed motor deficits in the beam test, without association with depressive or anxiety-like phenotype. However, *Dmd*^mdx^ rats exhibited a strong response to restraint-induced stress, with a large increase in freezings frequency and duration, suggesting an alteration in a functional circuit including the amygdala. In brain, large dystrophin isoform Dp427 was not expressed in mutant animals. *Dmd*^mdx^ rat is therefore a good animal model for preclinical evaluations of new treatments for DMD but care must be taken with their responses to mild stress.

## Introduction

Duchenne muscular dystrophy (DMD) is a X-linked neuromuscular disorder caused by mutations in the DMD gene, leading to a lack of dystrophin expression, a cytoskeletal protein mainly expressed in muscles, but also in other tissues like retina and brain. This disease is characterized by skeletal muscle pathology, but also cognitive and behavioral issues for around 20–50% of patients. Indeed, in addition to cognitive impairments [1], a subset of DMD patients suffer from attention-deficit/hyperactivity, anxiety, autism spectrum disorders, epilepsy and

**Competing interests:** The authors have declared that no competing interests exist.

obsessive-compulsive disorders [2–5]. The reasons explaining these impairments rely on the variable location of mutations in the DMD gene, affecting shorter brain dystrophin isoforms normally produced from independent promoters. The more severe cognitive impairments in patients are, the more distal part of this gene is affected with mutations [6]. As opposed to muscular symptoms, cognitive disabilities are not progressive, and not a consequence of muscle alterations. Cognitive functioning in DMD also includes deficits in linguistic functions [7], short- and long-term memories [7–9]. Impairments in different types of memories have been underlined in DMD patients, even with a normal IQ, suggesting a link with the full-length brain dystrophin commonly lost in all patients [10, 11]. In the brain, it is expressed in areas involved in cognition and emotional behavior, such as hippocampus, amygdala, cerebellum and sensory cortices. More precisely, those impairments seem to be related to the absence of dystrophin in hippocampal, cerebellar and prefrontal cortex synapses. In neurons, dystrophin selectively localizes to the postsynaptic membrane in inhibitory synapses and acts as an actin-binding postsynaptic scaffold in GABAergic synapses [12–15].

In the classical DMD model *mdx* mouse, the absence of the full-length brain dystrophin deficiency induces molecular, structural and physiological alterations in central inhibitory synapses, like an abnormal synaptic clustering and density of $GABA_A$ receptors in CA1 hippocampal dendritic layer [13, 16–18], thus facilitating NMDA receptor-dependent synaptic plasticity and also inducing an abnormally increased hippocampal LTP [19]. We have to note, as an aside, that t the clinical level, an abnormal distribution of $GABA_A$ receptors has also been found in brain of Duchenne patients [20]. In *mdx* mouse, long term object recognition memory is altered [21, 22], as well as the acquisition and long-term retention of fear memories, depending on the amygdala, and hippocampal-dependent learning strategy in the water maze [23]. However, no deficits are encountered in this model for spatial working memory, flexibility, perception and sensorimotor gating of auditory inputs [23]. This model of *mdx* mice is also known for their enhanced fearfulness [24]. Indeed, they display elevated levels of freezing behavior in response to mild behavioral stress or electric shock, in an independent way from skeletal muscle impairment, but dependent on brain dystrophin, as fear responses can be reduced by rescuing brain dystrophin expression [24, 25].

The role of dystrophin in the brain is still not fully understood. It is thought to have a role in executive functions, perception and information processing, but has not yet been extensively studied. In this study, we used the $Dmd^{mdx}$ rat model, which was recently generated [26] in order to counteract the minor clinical dysfunction of *mdx* mouse [27] and the fact that their small size imposes limitations in the analysis of several aspects of the disease. Moreover, rats display complex social traits and have a convenient size since they are 10 times larger than mice, allowing the possibility to collect large quantities of biological tissues compared to mice. But rats remain a small laboratory animal model and allow studies with high statistical power. In this model, muscular function has been investigated. We previously showed that at 3 months, forelimb, hindlimb, diaphragm and cardiac muscles displayed severe fiber necrosis. At 7 months, in skeletal muscles regeneration activity was decreased with muscle showing abundant peri- and endomysial fibrosis with some adipose tissue infiltration as in skeletal muscles from DMD patients. Furthermore, in $Dmd^{mdx}$ rats muscle, strength and spontaneous activity were decreased and fatigue was a prominent finding of muscle function analysis [26]. The purpose of this work is to characterize in detail locomotion parameters, anxiety behavior and freezing response of $Dmd^{mdx}$ rats emerging in response to a short restraint stress. Indeed, quantitative evaluation of fearfulness in animal models may provide a relevant readout in preclinical assessment of therapies targeting the skeletal but also central nervous system [28].

## Material and methods

### Animals

This study was approved by the Ethics Committee on Animal Experimentation of the Pays de la Loire Region, France, in accordance with the guidelines from the French National Research Council for the Care and Use of Laboratory Animals (Permit Numbers: APAFIS#10792–2017061316021120). All efforts were made to minimize suffering. Sprague-Dawley (SD/Crl) rats were obtained from Charles River (L'Arbresle, France) and $Dmd^{mdx}$ (KO) littermates animals of different ages were generated as previously described [26]. The rats were housed in a controlled environment (ventilated racks, ambient temperature of 21˚C, ambient hygrometry of 55%, 12 h light/dark cycle (dark at 8 pm, light at 8 am)), with several animals per cage, all males. Diet consisted of a standard diet (SAFE A04, Safe, Augy, France) given *ad libitum*, sterilized and filtrated water also given *ad libitum*. All behavioral tests were performed blind to animal identities.

### Ledged beam-walking test

$Dmd^{mdx}$ and WT animals aged 7 months were trained on a tapered/ledged beam-walking test, adapted from the procedure previously described [29]. This test is sensitive to dopaminergic function [30–32]. Rats walked along a 165 cm-long, progressively narrowing (6.5 cm wide at the wide end, 1.5 cm at the narrow end) Plexiglas beam, elevated above the floor on an incline of 15˚, to reach their home cage. Two cm below the beam was a 2.5 cm-wide Plexiglas ledge that provided a platform to step on when there was a motor deficit. This ledge allowed rats to express their motor deficit, and removed the need for postural compensation to prevent falling off the beam. Taking a step with only one or two toes on the main surface of the beam (and the other four or three toes overhanging the ledge) was scored as a half foot-fault, whereas stepping with the entire foot on the ledge rather than on the main surface of the beam was scored as a full foot-fault. Before testing, each animal was allowed one refresher trial, which was not video-taped. One test consisted of 3 consecutive trials videotaped from the rear to allow a clear observation of the hindlimbs.

### Elevated plus-maze

The elevated plus-maze is used to measure the level of anxiety-like behavior. The maze was made of 4 arms, two with high walls and two without walls and was elevated 1 meter above the floor, with an light intensity of 300 Lux in open arms. The behavioral model is based on the general aversion of rodents to open spaces. This aversion leads to thigmotaxis, a preference for remaining in enclosed spaces or close to the edges of a bounded space. In the elevated plus maze, this translates into the animals limiting their movement to the enclosed arms. At the start of each test, the rat was placed in the center of the maze, nose facing an open arm, and was allowed to explore it for 10 minutes. Each trial was videotaped, and the number of arm entries and the time spent in the opened and the closed arms were measured. The maze was cleaned with 70% ethanol between trials. Time spent in open arms was calculated, equal to (time spent in open arms) / (time spent in closed arms + time spent in open arms). This value is proportional to the anxiety level of the animal. Also, total number of entries into each arms, which is linked to global locomotion, was recorded.

### Sucrose preference

Sucrose preference is a locomotor-independent test in which the relative preference for a sucrose-sweetened solution (*vs*. water) gives a measure related to the anhedonia observed in

depressive patients [33]. Rats were given a choice between two bottles containing either tap water or 2% sucrose solution in their home cages during 48 h. The position of the two bottles was switched after 24 h in order to avoid a side preference. Water and sucrose consumptions were measured each day at the end of the afternoon by weighing the bottles. Sucrose preference (sucrose solution consumption (g) / water consumption (g) + sucrose solution consumption (g)) was calculated over the 48 h period and compared between both groups.

## Restraint-stress and open-field

6 weeks-old $Dmd^{mdx}$ and WT rats were weighed and then restrained for 10 seconds by grasping the scruff and back skin between thumb and other fingers, and tilting the animal upside-down in order that the ventral part of its body faced the experimenter. Immediately after they were placed in the center of the open-field arena (Bioseb) and recorded from above during 5 minutes, at a light intensity of 100 Lux. Non-stressed rats were removed from cage, weighed, and directly placed in the arena center.

Unconditioned fear responses induced by this short restraint stress were characterized by periods of tonic immobility (freezing) and quantified during the 5 min recording period. Complete immobilization of the rat, except for respiration, was regarded as a freezing response [34]. The percent time spent freezing was calculated.

## Central nervous system samples preparation for western blot studies

After euthanasia (deep anaesthesia and analgesia following i.p. injection of a mixture of ketamine (100 mg/kg, Imalgene, Merial, Lyon, France) and xylazine (10 mg/kg, Rompun, Bayer, Leverkusen, Germany) and then decapitation), brains from 6 weeks old animals from each genotype were extracted from the skull. For western blot analysis, samples from different brain regions were manually dissected, rapidly frozen in liquid nitrogen and stored at <-70˚C: medial prefrontal cortex (P), amygdala (A), dorsal hippocampus (D), ventral hippocampus (V), cerebellum (C). Finally, spinal cord (SC) was dissected, sampled as previously described [35], rapidly frozen in liquid nitrogen and stored at <-70˚C. Additionally, total brain (TB) samples consisting of a single brain hemisphere were prepared for each animal, and consisted of a whole brain preparation, with no distinction between brain areas.

## Central nervous system preparation for immunohistochemistry

For immunohistochemistry, whole brain from one to 3 months old animals was immediately cut into 5 slices at the level of respectively (i) the frontal pole, (ii) the optic chiasm, (iii) the oculomotor nerve, (iv) the midbrain and (v) the occipital pole. Each slice was immediately snap frozen in liquid nitrogen cooled isopentane and stored at <-70˚C. Brain slices were cut with a cryostat. Mouse monoclonal antibodies NCL-DYSB (1:25, Novocastra Laboratories (NL), Newcastle, UK), MANEX5556 (1:50, Developmental Studies Hybridoma Bank, University of Iowa, Iowa City, IA) and NCL-DYS2 (1:50, NL) were respectively used for the detection of Dp427, Dp427/Dp140 and Dp427/Dp140/Dp71 isoforms. For DYSB only, transverse cryosections were placed in 0.01 M citric acid, 0.05% Tween 20 (pH 6) and placed in a water bath for 15 min at 98˚C. For all immunolabellings, sections were pre-incubated in PBS with 5% normal goat serum (Dako) for 30 min at room temperature (RT) and then incubated with primary antibody in 5% rat serum overnight at 4˚C. After washing, primary antibody was revealed using a biotinylated secondary antibodies (1:300, Dako) in PBS with 5% rat serum for 1 hour at RT. Bound antibodies were detected with streptavidin (Dako) and DAB Liquid Substrate (Dako) for immunoperoxidase. Slides were counterstained with Gill's hematoxylin and mounted. All slides evaluations were performed by a skilled pathologist certified by the European College of Veterinary Pathology.

## Western blotting

Total proteins from different brain areas were extracted using 400 μL of RIPA extraction buffer containing protease inhibitors (Roche) and ground with TissueLyser II (Qiagen). 80 μg (for MANEX 1011C antibody) or 30 μg (for NCL-DYS2 antibody) of protein extracts were loaded on a 3–8% Tris-Acetate Precast polyacrylamide gel of NuPAGE Large Protein blotting kit (Invitrogen). Additionally, controls were loaded on each gel: muscle protein extract (*biceps femoris*) from a WT rat for detection of Dp427 and total brain protein extract for Dp71 and Dp140. In order to compare dystrophin levels between brain areas, these samples were loaded on one unique gel for the same animal. After Red Ponceau staining, membranes were incubated with two different mouse anti-Dystrophin antibodies: NCL-DYS2 (1:100, Novocastra), for the detection of Dp71 and Dp140 isoforms, MANEX 1011C (1:250, MDA Monoclonal Antibody Resource) for the detection of Dp427 isoform. An anti-GAPDH antibody (1:10000, Imgenex) was also used as a loading control. Detection was performed using either a secondary anti-mouse IgG HRP-conjugated antibody P0447 (1:5000, Dako) for dystrophin primary antibodies or secondary anti-goat IgG HRP-conjugated antibody P0449 (1:2000, Dako) for GAPDH primary antibody. Immunoblots were revealed with ECL Western blotting substrate (Pierce) and exposed to ECL-Hyperfilm (Amersham). For the semi-quantification analysis of the data, levels of dystrophin isoforms in each central nervous system region (n = 4 per genotype) were first normalized to GAPDH levels using ImageJ software, and then values were normalized to dystrophin Dp427 control muscle levels or to total brain Dp71 and Dp140 levels.

A schematic representation of the dystrophin epitopes recognized by all antibodies used in this study is depicted in Fig 1 (Fig 1).

## Data analysis

All values are expressed as mean ± SEM, with a significance level set at p < 0.05. Statistical evaluation has been performed by using GraphPad Prism 5 (GraphPad Software, Inc., La Jolla, CA, USA). Data distribution was first evaluated with a D'Agostino & Pearson omnibus normality test, for each group, at each tested timepoint, for each measured parameter. We then used either Kruskal Wallis test or ordinary two-ANOVA analysis and if a significant interaction was found, we further performed multiple comparison tests.

## Results

### *Dmd^mdx* rats display neuromotor alterations in the ledged beam-walking test

In order to complete the assessment of locomotor functions of *Dmd^mdx* animals, as we did in our previous study [26], we used the ledged beam-walking test at 7 months, a time point where the muscular phenotype is strongly affected in *Dmd^mdx* rats. Time spent to cross the beam was compared between WT and *Dmd^mdx* rats. The latter animals spent more time crossing the apparatus compared to WT controls (p = 0.0339, Fig 2A), and this was correlated with a non-significant increase in the total number of steps to cross the beam (p = 0.1229, Fig 2B). However, no impairment was found for *Dmd^mdx* rats in terms of stepping errors when performing the task (front limbs: p = 0.2333, hind limbs: p = 0.6378, Fig 2C).

### Anhedonia in the sucrose preference test and anxiety levels in the elevated plus maze are not affected in *Dmd^mdx* rats

The sucrose preference test is usually used to evaluate anhedonia, indicated by a decrease in sucrose consumption, a typical depression-like behavior in rats and mice. Here evaluated this

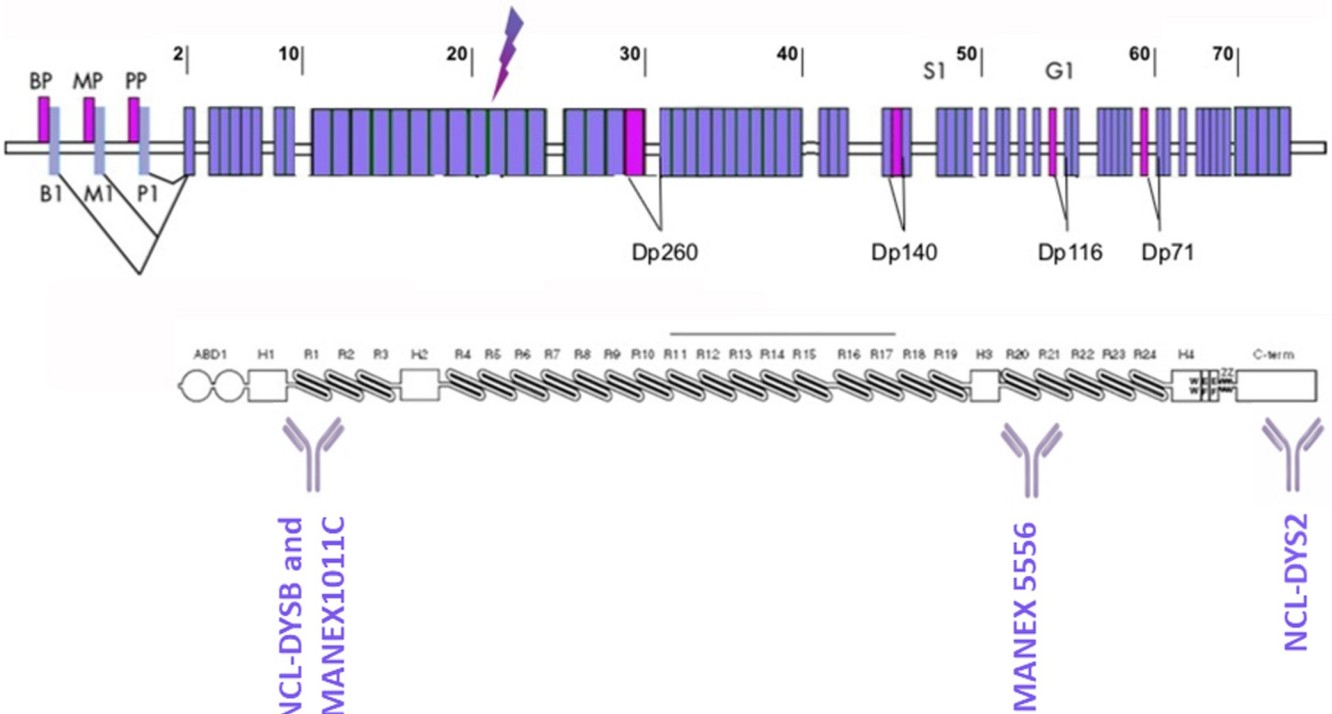

**Fig 1. Dystrophin epitopes recognized by different antibodies used in the study.** Western-blot antibodies (NCL-DYS2 and MANEX1011C) and immunolabelling antibodies (MANEX5556 and NCL-DYSB) are represented. Letters indicate the various promoters: brain (B), muscle (M), and Purkinje (P) promoters, and different isoforms Dp260, Dp140, Dp116 and Dp71 are represented. The arrow indicates the mutation position in $Dmd^{mdx}$ rat model.

parameter at 2 different ages in terms of disease progression, 3 and 7 months. At 3 months and 7 months, no age nor genotype effect on sucrose preference was found, indicating that $Dmd^{mdx}$ rats had the same level of preference that WT rats over time (age: $F(1,20) = 0.0125$, $p = 0.912$; genotype: $F(1,20) = 2.537$, $p = 0.127$, Fig 3A), despite a large variability for older $Dmd^{mdx}$ animals, with some of them having a very low sucrose preference. In parallel, water consumption

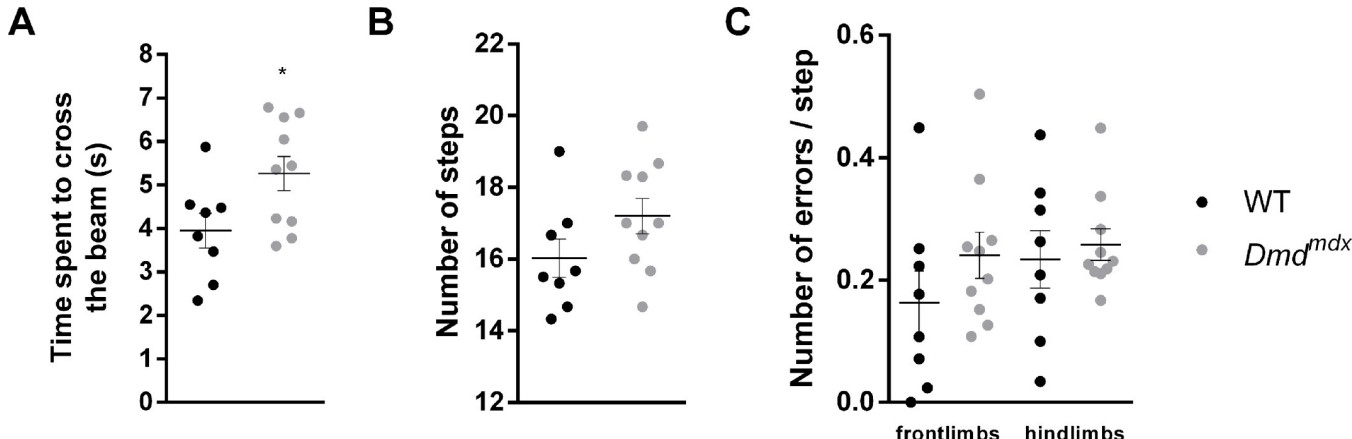

**Fig 2. Motor functions impairment in the ledged-beam walking test in $Dmd^{mdx}$ rats at 7 months.** For WT (black dots, n = 8) and $Dmd^{mdx}$ rats (grey dots, n = 10), (A) Time spent to cross the beam was significantly higher in mutant animals, (B) Number of steps was also higher, but without reaching significance, and (C) Number of errors per step was unaffected. Data are expressed as mean ± SEM. $^*p < 0.05$.

was evaluated and was shown to be constant over age and between both genotypes (age: $F_{(1,20)}$ = 0.825, p = 0.374; genotype: $F_{(1,20)}$ = 0.894, p = 0.356, Fig 3B). In order to study whether anxiety behavior is affected in $Dmd^{mdx}$ rats, we used the elevated plus-maze test. The percentage of time spent in the open arms, indicative of the anxiety level of the animal, was not significantly different between WT and $Dmd^{mdx}$ rats (age: $F_{(1,20)}$ = 2.474, p = 0.131; genotype: $F_{(1,20)}$ = 0.568, p = 0.460, Fig 3C), despite the fact that the total number of entries into either open or closed arms was significantly lower for $Dmd^{mdx}$ rats at both ages, due to locomotor impairment (age: $F_{(1,20)}$ = 25.49, p < 0.0001; genotype: $F_{(1,20)}$ = 26.38, p < 0.0001; Sidak's multiple comparison test, p < 0.01 at 3 months and 7 months, Fig 3D). These results suggest that in this test, even if locomotion is affected, $Dmd^{mdx}$ animals are not more anxious than WT animals.

## Behavioral response to restraint-induced mild stress

Locomotion was studied in an open-field arena for younger (6 weeks old) $Dmd^{mdx}$ mutants and WT rats. We used this age in order to assess if the response to stress is affected at an early

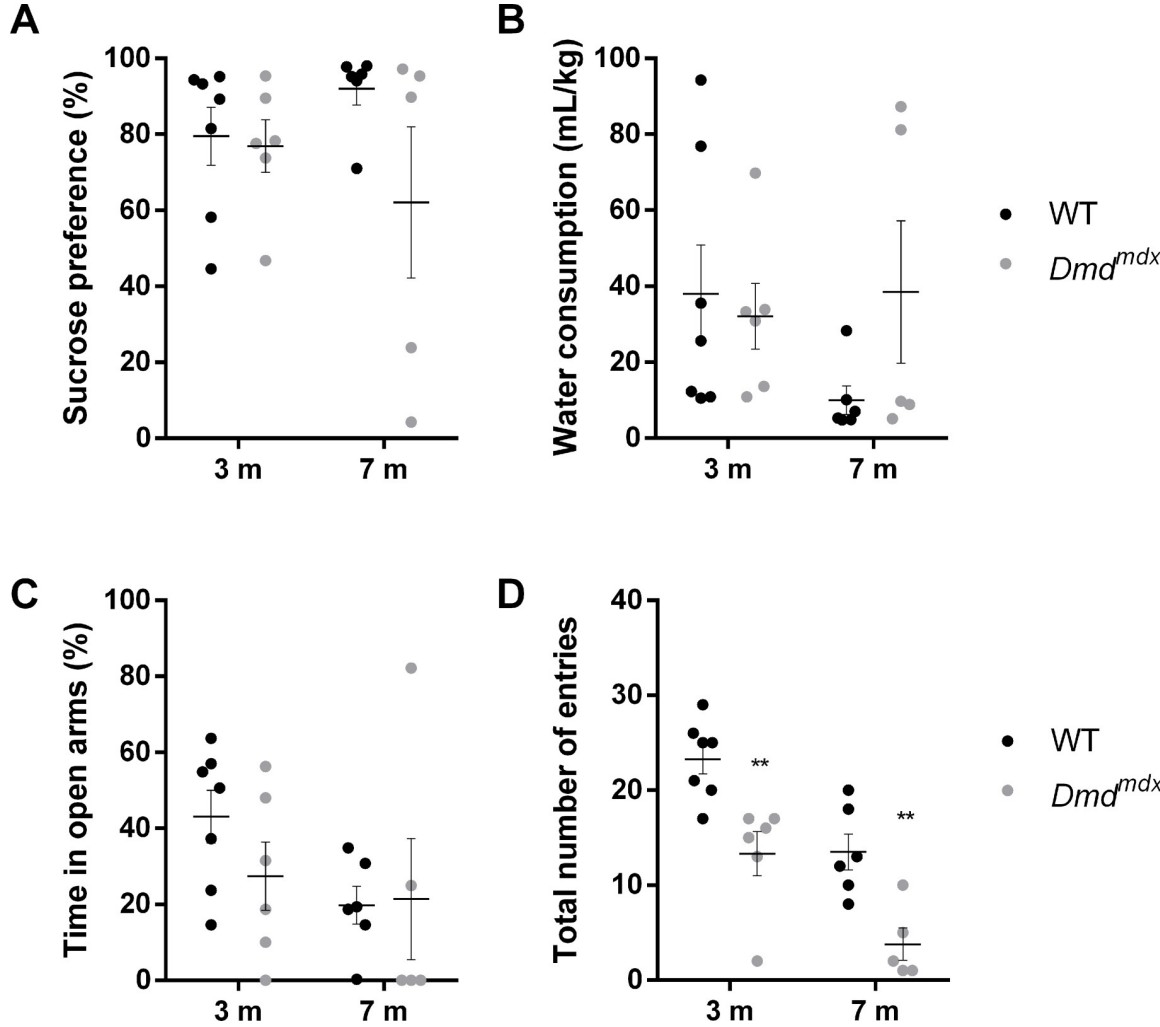

**Fig 3. Anhedonia and anxiety are not affected in $Dmd^{mdx}$ rats at 3 and 7 months.** For WT (black dots, n = 7) and $Dmd^{mdx}$ rats (grey dots, n = 6), (A) Sucrose preference and (B) Water consumption are not affected in dystrophin-deficient rats, neither at 3 months, nor at 7 months, (C) Time spent in the open arm of the elevated plus maze (% of time spent on open and closed arms) is also stable over time in both genotypes, (D) Total number of entries into open and closed arms is significantly lower in $Dmd^{mdx}$ rats. Data are expressed as mean ± SEM. **p < 0.01.

timepoint. Duration of stress-induced tonic immobility, considered as a measure of unconditioned fearfulness [24], was also analyzed. For the retention group, animals were gently scruff-restrained for 10 seconds, in a way similar to the one used for the immobilization of rats for standard examination or intraperitoneal injections, whereas control animals were directly placed inside the open-field arena. *Dmd^{mdx}* rats walked a shorter distance that WT littermates, and when looking at the effect of retention stress, we found that it only had an effect on *Dmd^{mdx}* animals, because mutants from the retention group were less mobile than the unstressed *Dmd^{mdx}* and WT animals (stress: $F_{(1,52)} = 3.9$, $p = 0.0536$; genotype: $F_{(1,52)} = 54.38$, $p < 0.0001$, Tukey's multiple comparison test, $p < 0.05$ for *Dmd^{mdx}* control vs. *Dmd^{mdx}* retention, Fig 4A). However no impact of retention was found on mean speed during the test, a genotype effect was detected (stress: $F_{(1,52)} = 0.742$, $p = 0.393$; genotype: $F_{(1,52)} = 29.3$, $p < 0.0001$, Fig 4B). All animals from each group had the same levels of anxiety, as indicated by similar levels of thigmotaxis, with no effect of genotype nor stress (stress: $F_{(1,52)} = 0.259$, $p = 0.613$; genotype: $F_{(1,52)} = 0.746$, $p = 0.392$, Fig 4C). Interestingly, we show that restraint induced a large increase in tonic immobility in *Dmd^{mdx}* rats characterized by a lasting freezing like behavior in terms of freezing duration (+83% between both *Dmd^{mdx}* groups, stress: $F_{(1,52)} = 16.22$, $p = 0.0002$; genotype: $F_{(1,52)} = 49.83$, $p < 0.0001$, Tukey's multiple comparison test, $p < 0.0001$ for *Dmd^{mdx}* control vs. *Dmd^{mdx}* retention, Fig 4D) and number of freezings

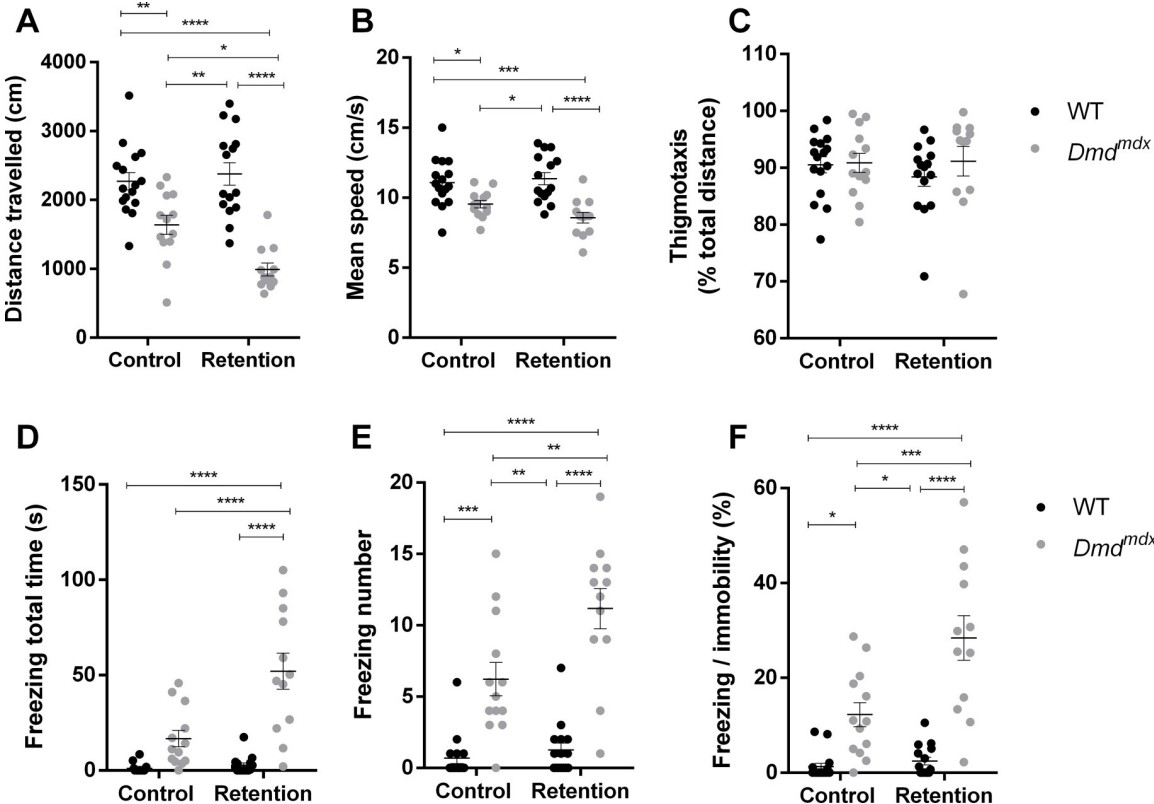

**Fig 4. *Dmd^{mdx}* rats are characterized by early locomotion deficits (6 weeks old) and a maladaptive response to mild stress.** Compared to WT littermate controls (black dots, n = 16 unstressed and n = 15 stressed), *Dmd^{mdx}* rats (grey dots, n = 13 unstressed and n = 12 stressed), stressed and non-stressed were characterized by a lower travelled distance (A) and mean speed (B) in the open-field, without effect on thigmotaxis (C). Specifically, stress significantly induced higher total freezing time (D) and number (E) and degree of freezing compared to immobility (F) in *Dmd^{mdx}* animals. Data are expressed as mean ± SEM. *$p < 0.05$, **$p < 0.01$, ***$p < 0.001$, ****$p < 0.0001$.

(+80% between both $Dmd^{mdx}$ groups, stress: F(1,52) = 9.75, p = 0.0029; genotype: F(1,52) = 76.47, p < 0.0001, Tukey's multiple comparison test, p < 0.01 for $Dmd^{mdx}$ control vs. $Dmd^{mdx}$ retention, Fig 4E). When freezing behavior was normalized to basal immobility, the same significant effect of stress was found throughout the 5-min testing period (stress: F(1,52) = 12.82, p = 0.0008; genotype: F(1,52) = 58.24, p < 0.0001, Tukey's multiple comparison test, p < 0.001 for $Dmd^{mdx}$ control vs. $Dmd^{mdx}$ retention, Fig 4F). These results indicate that $Dmd^{mdx}$ animals, as opposed to WT littermates, display a maladaptive response to moderate stress, without being more anxious. These performances in the open-field cannot be attributed to overt changes in the behavioral expression of the fear response, as excessive grooming behavior was not observed during the sessions (stress: F(1,52) = 0.215, p = 0.645; genotype: F(1,52) = 0.276, p = 0.602, Fig 5A). Finally, vertical activity was only impacted by genotype, confirming the motor impairment, but not impacted by stress retention (stress: F(1,52) = 0.919, p = 0.342; genotype: F(1,52) = 33.60, p < 0.0001, Fig 5B).

## Dp71, Dp 140 and Dp427 levels in different regions of central nervous system

Western blot analysis of dystrophin levels and immunolabelling of dystrophin in different areas of central nervous system were performed using two antibodies directed against epitopes located at the C-terminus (for Dp71 and Dp140 detection) and within exons 10–11 (for full dystrophin Dp427 detection). A normalization step was added by using total WT rat brain for Dp71 and Dp140, because these isoforms are absent from WT muscles, and by using WT skeletal muscle extract for Dp427 quantification, this isoform being highly expressed in muscle.

Using western blot in WT animals, Dp71isoform, implicated in transmembrane receptor binding and vascular development, was found equally distributed throughout brain areas and spinal cord, as expected. Due to the point mutation in exon 23, $Dmd^{mdx}$ and WT animals

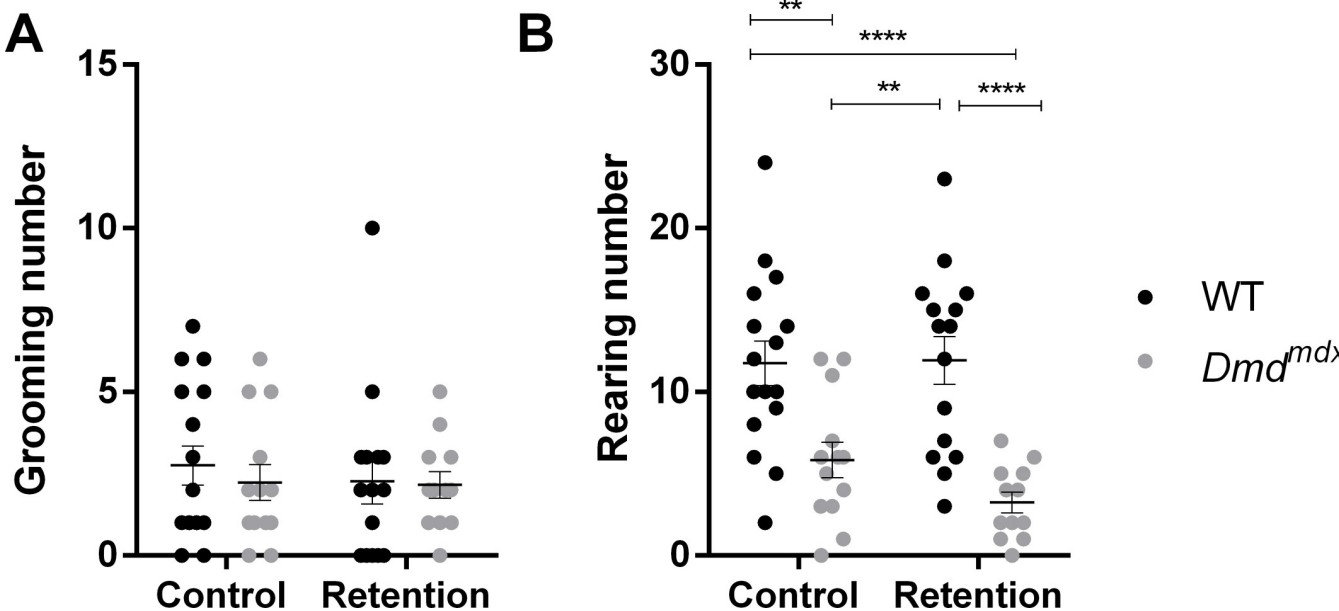

**Fig 5. Mild stress has no specific effect on ethological parameters grooming and rearings in young $Dmd^{mdx}$ rats (6 weeks old).** For WT (black dots, n = 16 unstressed and n = 15 stressed) and $Dmd^{mdx}$ rats (grey dots, n = 13 unstressed and n = 12 stressed), (A) Grooming number is stable in both genotypes, with or without prior stress, (B) Vertical activity, measured by rearing number is also unaffected by stress, but confirms the locomotor deficit in mutant rats. Data are expressed as mean ± SEM. **p < 0.01, ****p < 0.0001.

displayed the same absence of regional specificity of Dp71 expression in central nervous system (genotype: $F_{(1,42)}$ = 1.526, p = 0.224; region: $F_{(6,42)}$ = 1.218, p = 0.316, Fig 6A and 6B). Concerning Dp140 isoform, which participates to early development via regulation of neuron differentiation, neuron projection morphogenesis and chromatin modification, we found in WT animals the highest levels in cerebellum, which happen to be significantly higher than amygdala levels. Dp140 levels in the same areas from *Dmd*<sup>mdx</sup> animals were not different than WT littermates (genotype: $F_{(1,42)}$ = 3.944, p = 0.054; region: $F_{(6,42)}$ = 11.99, p < 0.0001, Fig 6A and 6C). Interestingly, Dp140 was found at very low levels in spinal cord of animals from both groups, compared to other brain areas. Finally, we found that Dp427 isoform, which is found to the synaptic membrane of neurons with a function revolving around transmembrane transport and signal transmission, was unequally distributed throughout central nervous system areas of WT rats, with higher levels in cerebellum and spinal cord (genotype: $F_{(1,42)}$ = 73.90, p < 0.0001; region: $F_{(6,42)}$ = 74.24, p < 0.0001, Fig 6A and 6D). This is in line with previous mouse studies [8], where this isoform was found to be expressed in the cerebellum. As

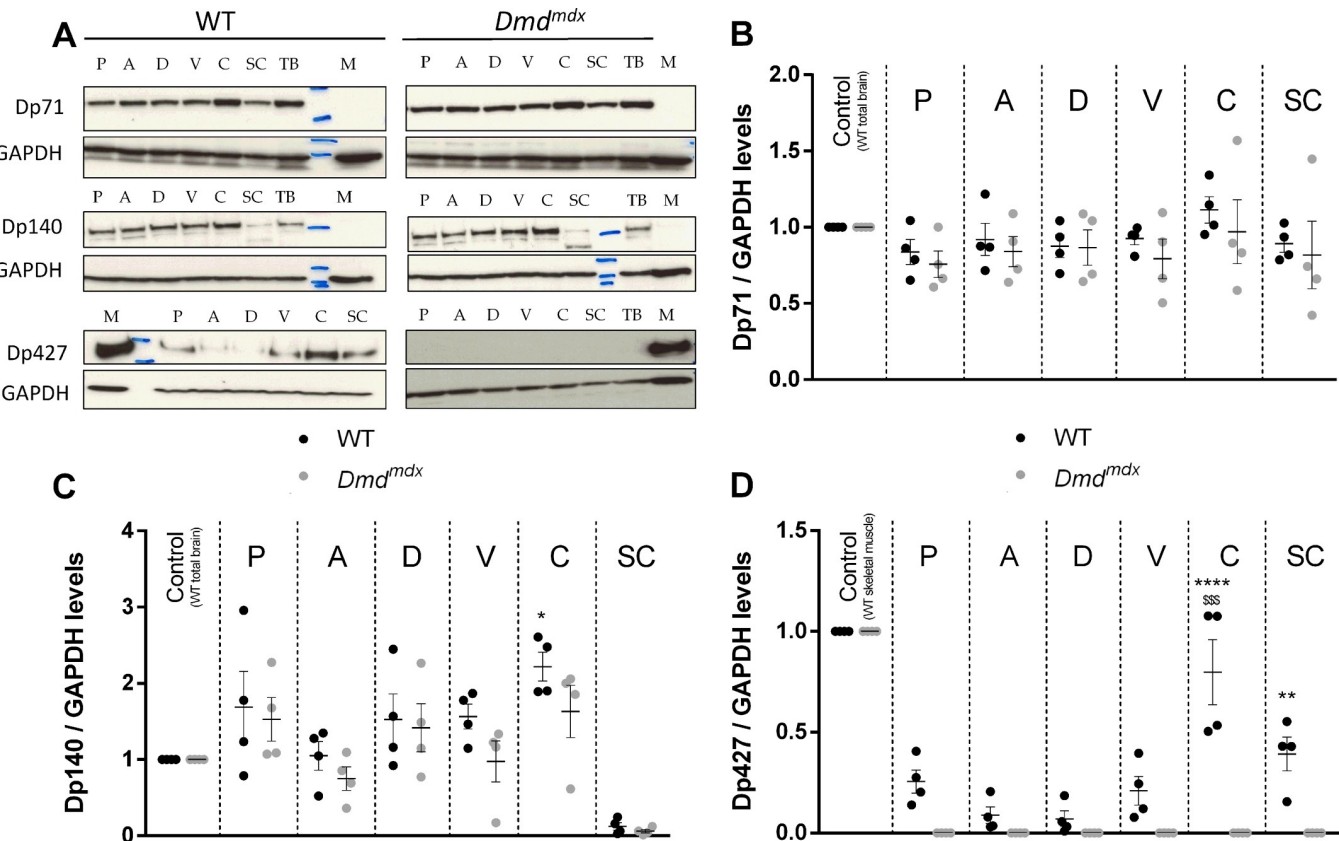

**Fig 6. Specific lack of Dp427 dystrophin isoform in *Dmd*<sup>mdx</sup> rats central nervous system, but no modification of Dp140 and Dp71 levels.** WT controls (black dots, n = 4) and *Dmd*<sup>mdx</sup> animals (grey dots, n = 4) were sacrificed and samples from brain areas (P: prefrontal cortex, A: amygdala, D: dorsal hippocampus, V: ventral hippocampus, C: cerebellum), spinal cord (SC), total brain (TB) with no specific targeted area and *biceps femoris* muscle from a WT rat (M), were harvested. Western-blot of total proteins was incubated with either monoclonal antibodies NCL-DYS2 and 30 μg protein loading for Dp71 and Dp140) or Manex1011C and 80 μg protein loading for Dp427 (A). Quantification on western-blot of dystrophin isoforms in each central nervous system region (n = 4 per genotype) were normalized to GAPDH levels, and then to dystrophin Dp427 control muscle levels (B, C). This revealed identical levels of 71 kDa (A, B) and 140 kDa (A, C) dystrophin isoforms in all studied central nervous system areas from WT and *Dmd*<sup>mdx</sup> animals. Only Dp140 cerebellar levels were significantly higher compared to amygdalar levels in WT rats. Dp427 isoform was not detected in *Dmd*<sup>mdx</sup> animals (A, D), and had variable levels in WT rats, with higher levels in cerebellum and spinal cord. Total WT brain extracts were used as positive controls for Dp71 and Dp140 detections, and *biceps femoris* muscle from a WT rat was used as positive control for Dp427 detection. Staining with an anti-GAPDH polyclonal antibody validated equal protein loadings. Data are expressed as mean ± SEM. In Fig 6C, *p < 0.05 vs. A. In Fig 6D, ****p < 0.0001 vs. SC, $^{\$\$\$}$p < 0.001 vs P, A, D and V and **p<0.01 vs. A and D.

anticipated, using western blot no Dp427 was found for *Dmd^mdx* rats, confirming the effectiveness of the generated mutation.

After immunohistolabelling using DYSB antibody, Dp427 was observed in WT animals with a faint intensity in large neuronal bodies of the hippocampal and cerebellar areas of the brain (Fig 7). This isoform was absent from the *Dmd^mdx* littermates. Immunolabelling of Dp427/Dp140 isoforms, using MANEX5556 antibody, was similar in WT and *Dmd^mdx* animals, therefore indicating that this marking is probably Dp140 specific, as we showed before that Dp427 is absent from mutant animals. This isoform was found with a light intensity in smooth muscle layer of blood vessels present in the whole brain area, but we cannot exclude the presence of a glial marking around vessels. Lastly, immunolabelling of Dp427/Dp140/Dp71 isoforms was found with similar intensities in WT and *Dmd^mdx*, therefore indicating that this marking is probably Dp71 specific, as we showed before that Dp427 is absent from mutant animals and that Dp140 is only visible smooth muscle layers. Dp71 isoform was found exclusively in endothelial cells of blood capillaries throughout the entire brain area. No associated histologic lesions were identified.

## Discussion

This study describes the neuromotor, anxiety and anhedonia characterization of the recently described dystrophin deficient rat model, *Dmd^mdx* rats [26]. It was previously demonstrated that cardiac and skeletal phenotypic properties of this model are very close to the human DMD pathology. Duchenne muscular dystrophy is the most common neuromuscular disorder, representing about 30% of muscular dystrophies [6, 36, 37], without any curative treatment to date. The present results demonstrate that *Dmd^mdx* rats display neuromotor alterations at 7 months, as shown with the transversal beam test, in which mutant animals spent more time to cross the beam. However, no significant changes were found in the number of errors, neither for frontlimbs nor hindlimbs. This function mainly relies on the nigrostriatal dopaminergic pathway, which is known to control the dexterity of movement [38], correlated with the number of errors performed upon crossing the beam. Locomotor measurements are important parameters allowing to define phenotypes of animal models with muscular dystrophies. Motor deficit underlined here may therefore be a consequence of dystrophin absence in muscles from the limbs, but we cannot exclude they may also result from reduced levels of cerebellar dystrophin, as demonstrated by western blot.

In DMD patients, clinically significant internalizing disorders including anxiety and depression have been demonstrated, so that even a Duchenne muscular dystrophy neuropsychiatric syndrome has been suggested [39]. Indeed, a specific study on DMD patients demonstrated that 24% and 19% of them displayed anxiety and depression disorders, respectively [40]. Therefore, in order to measure anhedonia in mutant *Dmd^mdx* animals, we used the sucrose preference test, reflecting the hedonic drive in a locomotor-independent manner [41]. The preference for sucrose solution was found to be unmodified in 3 months animals and not significantly reduced in 7 months old animals lacking dystrophin. This further supports the idea that this model does not manifest any depressive-like phenotype. In *Dmd^mdx* rats, sucrose preference appears to decrease in a non-significant manner over time, this could be linked to the behavior of some animals that did not show any interest for the sucrose solution. Moreover, *Dmd^mdx* rats performances in the elevated plus maze display a high variability indicating that anxiety-like behavior is absent in this model, as confirmed by the same level of thigmotaxis and grooming behavior in the open-field test between both groups. In this study, we have to report that motor impairment revealed in the beam test might have biased the assessment of emotional behaviors, which in many tests with a high-motor demand is dependent on

WT    *Dmd^mdx*

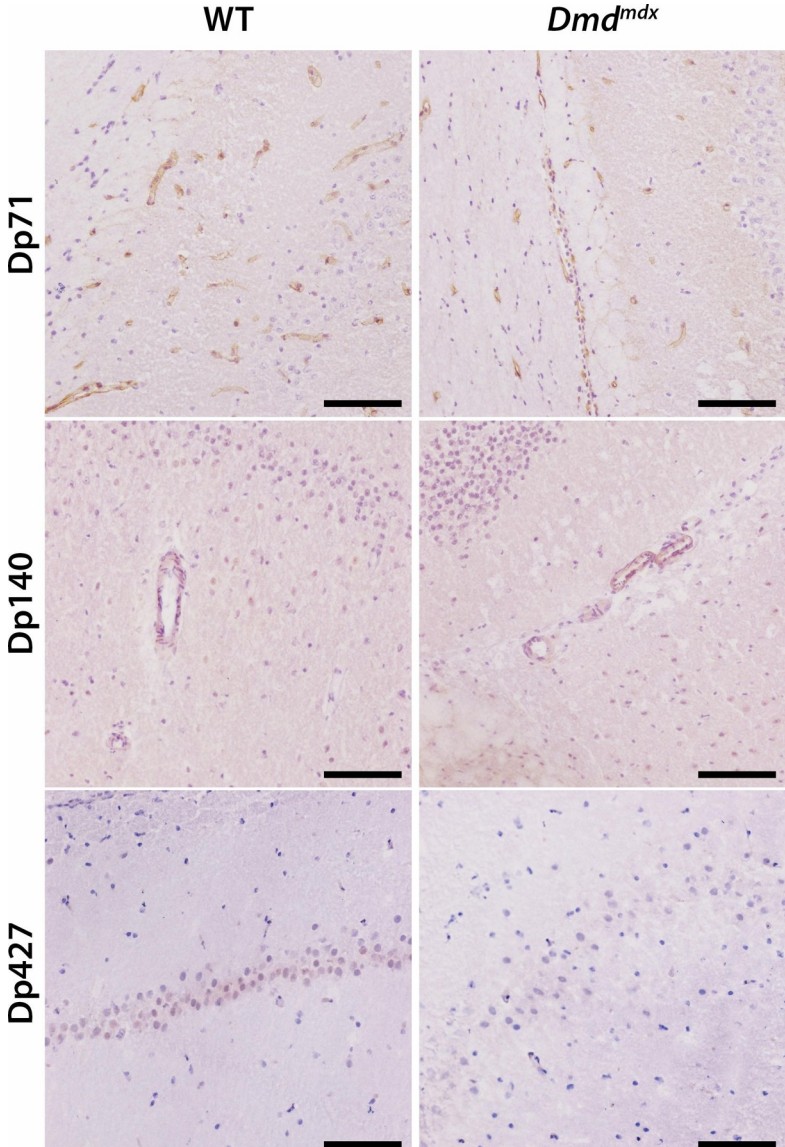

**Fig 7. Lack of Dp427 dystrophin isoform in *Dmd^mdx* rats brain and no modification in distribution of Dp140 and Dp71 isoforms.** Males WT controls (n = 3) and *Dmd^mdx* rats (n = 3) were sacrificed and slides from brain areas (at the levels of the frontal pole, the optic chiasm, the oculomotor nerve, the midbrain and the occipital pole) were processed for immunolabelling of Dp71, Dp140 and Dp427 dystrophin isoforms using respectively monoclonal antibodies NCL-DYS2, MANEX5556 and NCL-DYSB. Isoforms Dp71 and Dp140 were detected with similar intensities in WT littermate controls and *Dmd^mdx* rats and were respectively localized in endothelial cells of blood capillaries and in smooth muscle layer of blood vessels throughout brain parenchyma. Isoform Dp427 was mostly detected in the cytoplasm of some large neurons of the hippocampus in WT controls but was absent in *Dmd^mdx* littermate rats.

mobility. Indeed, total number of entries into the elevated plus maze arms, an indicator of global locomotion, is significantly reduced at each studied timepoint. However, time in open arms did not differ between groups, so no misleading interpretation could be made. Contrasting results have been found on anxiety in the classical model of *mdx* mice. Indeed, in 2009 a group showed that this model did not display an anxiety-like phenotype in the elevated plus maze [24]. However, other recent studies indicate a deficit in the light-dark choice anxiety test [25, 42]. Moreover, another group demonstrated anxiety-like and depression-like behaviors in *mdx* mice, associated with decreased BDNF (Brain derived neurotrophic factor) levels [43]. It

may therefore be useful to evaluate BDNF levels and perform light-dark choice anxiety tests to fully characterise the anxiety phenotype in $Dmd^{mdx}$ rats.

The open-field test was used to assess horizontal (distance and speed) and vertical (rearing) activities, which were shown to be reduced in 6 weeks-old $Dmd^{mdx}$ rats compared to WT rats. This reduced locomotion in the open-field arena has been previously reported for $Dmd^{mdx}$ animals [26], as opposed to studies using *mdx* mice, which do not display any motor deficit before 6 months of age [24, 44], thus confirming the early locomotor deficit in this rat DMD model. It was previously shown that the rodent normal defensive behavior in response to danger or a threat is enhanced in *mdx* mice, showing potent defensive freezing responses to a short stress restraint, as opposed to WT animals, in a way independent from hypothalamic–pituitary–adrenal axis activation [24]. Here we show for the first time that a 10 seconds restraint stress on young $Dmd^{mdx}$ rats has an effect on exploratory behavior. Indeed, freezing, defined as a lasting tonic immobility involving a brain circuit including the amygdala [24], is increased by retention for $Dmd^{mdx}$ rats in terms of freezing time and number of freezing events, while this effect is absent on WT littermates. It is important to notice that this effect is also found at a lower level in non-stressed $Dmd^{mdx}$ animals compared to WT. Therefore we have an exacerbation of freezing behavior following a brief stress event. This maladaptive response to stress confirms freezing behavior as an important and reliable study parameter in $Dmd^{mdx}$ rat. These analyses were performed on young animals at the age of 6 weeks, and we cannot exclude that this phenotype will be exacerbated at later timepoints, which has to be taken into account in the design and follow-up of preclinical studies using this model. In *mdx* mice, this behavior has been shown to increase with age and is thought to be underlined by an alteration of amygdala GABAergic circuits in dystrophin deficient animals [25]. We may hypothesize the same thing is occurring in our model, as well as alterations of central serotonin [44, 45] and cholinergic functions [46]. Reduced mobility of dystrophin deficient animals is often attributed to muscle wasting inducing fatigue [47]. Here we show that for $Dmd^{mdx}$ rats, this should be interpreted with care because this reduced mobility may indeed be a result of higher stress reactivity. For instance, when performing an intra-peritoneal injection, scruff restraint might appear to be a relatively mild stress, but in fact leading to confounding effects on locomotion parameters often used to evaluate treatments efficacies in preclinical studies. Indeed, this stress response is characterized by a strong motor inhibition which may be critical in the interpretation of functional measures based on quantification of mobility. One might hypothesize that this higher stress response in $Dmd^{mdx}$ rats is due a higher pain sensitivity, but during the evaluation animals did not show any paw rigidity while being in a tonic immobility state, and they reacted to sudden noises occurring in the animal house or touching, therefore interrupting their freezing behavior.

To understand the molecular mechanisms underlying these behavioral deficits, we performed studies of dystrophin isoforms expression level and localization in different brain areas. As demonstrated with the immunohistolabelling study, Dp427 isoform was present in large neuronal bodies in WT rats, and we found an absence of Dp427 isoform in all studied regions from $Dmd^{mdx}$ rats, which is consistent with the mutation generated on exon 23, inducing a complete loss of this large isoform and indicates that $Dmd^{mdx}$ rats are indeed dystrophin deficient animals, in muscles as well as in central nervous system. In WT animals, the highest levels of Dp427 isoforms were detected in the cerebellum and spinal cord, which is in agreement with studies demonstrating strong dystrophin expression in cerebellum, but also cortex and hippocampus in WT mice and rats [48, 49]. As prefrontal cortex and amygdala are known to communicate and be both involved in stress response [50, 51], dystrophin levels in these brain areas might participate in a typical adaptive stress response for WT rats. It was previously shown that brain dystrophin Dp427 is associated with a subpopulation of $GABA_A$ receptors at

inhibitory synapses, and that in *mdx* mice, an impaired clustering of $GABA_A$ receptors in hippocampus, amygdala and cerebellum [12, 13, 16, 24, 26] has been associated with altered synaptic plasticity [22, 24, 52, 53]. Therefore, brain dystrophin deficiency in $Dmd^{mdx}$ rats may also affect synaptic transmission and GABAergic communication between cortex and amygdala, and more generally, the integrity of several brain areas might be compromised. Concerning the hippocampus, spatial memory needs to be evaluated in this model in order to assess potential specific short-term memory deficits. The known functional segmentation between dorsal hippocampus and ventral hippocampus, with the ventral part being involved with emotion and the dorsal part regulating information processing [54, 55] is also found here with dystrophin levels that seem to be higher in the ventral hippocampus. Behavioral and physiological consequences of this observation still need to be evaluated. Other structures expressing Dp427 in WT animals are implicated in stress response, like entorhinal and perirhinal cortices, hippocampus or cerebellum [56], so the dystrophin deficiency induced in these regions in our model may contribute to the abnormal contention response and to the locomotor deficits detailed in this study. Additionally, levels of other dystrophin isoforms Dp140, expressed in smooth muscle layer of blood vessels and Dp71, exclusively expressed in endothelial cells of blood capillaries from both groups, are not impacted by the dystrophin-deficient phenotype in comparison to control animals.

Taken together, these findings provide insights into the relevance of using the $Dmd^{mdx}$ rat model, with freezing confirmed as a good readout to assess dynamics of therapeutics targeting brain functions in DMD models. Moreover, given the robust clinical relationship between muscular and locomotion deficits already demonstrated in this model and human Duchenne pathology, present data further emphasize a potential beneficial role of $Dmd^{mdx}$ rat model to better understand human central nervous system symptoms to stress conditions in a Duchenne disease context.

## Supporting information

**S1 Raw images.**
(PDF)

## Acknowledgments

We thank people from technical team of the animal house IRS2, who took care of all animals throughout these studies. We also thank the MDA Monoclonal Antibody Resource for providing the MANEX 1011C antibody.

## Author Contributions

**Conceptualization:** Dorian Caudal, Corinne Huchet.

**Data curation:** Dorian Caudal.

**Formal analysis:** Dorian Caudal, Thibaut Larcher.

**Methodology:** Dorian Caudal, Virginie François, Aude Lafoux, Mireille Ledevin, Thibaut Larcher.

**Validation:** Corinne Huchet.

**Writing – original draft:** Dorian Caudal.

**Writing – review & editing:** Ignacio Anegon, Caroline Le Guiner, Thibaut Larcher, Corinne Huchet.

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
