## [Decision Letter · Decision Letter 0]

2 Jan 2020

PONE-D-19-28211

Dystrophin-deficient Dmdmdx rat model displays an increased behavioral response to restraint-induced mild stress

PLOS ONE

Dear Dr. Caudal,

Thank you for submitting your manuscript to PLOS ONE. After careful consideration, we feel that it has merit but does not fully meet PLOS ONE’s publication criteria as it currently stands. Therefore, we invite you to submit a revised version of the manuscript that addresses the points raised during the review process.

Your manuscript has been evaluated by 2 reviewers. One of them raised several major and some minor concerns that need to be addressed and consequently a major revision is suggested. A few more technical points were  raised by the second referee. However, there was also some concern about one of the assays being underpowered.

We would appreciate receiving your revised manuscript within the next 6 weeks. To enhance the reproducibility of your results, we recommend that if applicable you deposit your laboratory protocols in protocols.io, where a protocol can be assigned its own identifier (DOI) such that it can be cited independently in the future. For instructions see: http://journals.plos.org/plosone/s/submission-guidelines#loc-laboratory-protocols

We look forward to receiving your revised manuscript.

Kind regards,

Gerhard Wiche, Ph.D.

Academic Editor

PLOS ONE

Journal Requirements:

2. Please amend the methods section of your manuscript to include the method of euthanasia

Reviewers' comments:

Reviewer's Responses to Questions

**Comments to the Author**

1. Is the manuscript technically sound, and do the data support the conclusions?

Reviewer #1: Partly

Reviewer #2: Yes

2. Has the statistical analysis been performed appropriately and rigorously? 

Reviewer #1: Yes

Reviewer #2: Yes

3. Have the authors made all data underlying the findings in their manuscript fully available?

Reviewer #1: Yes

Reviewer #2: Yes

4. Is the manuscript presented in an intelligible fashion and written in standard English?

Reviewer #1: Yes

Reviewer #2: Yes

5. Review Comments to the Author

Reviewer #1: This work by Caudal et al focuses on the behavioral characterization of the dystrophin deficient DMD rat model previously described by some of the authors.

The manuscript is relatively well written, with experiments and results well designed and easy to follow. The figures are also very clear. Overall, this is an interesting study which confirms a behavioral phenotype observed in another animal model of DMD, the mdx mouse and is therefore of interest to the research community in the DMD field.

However, the authors draw conclusions that may be premature based on the presented results. Some experiments do not strongly support the conclusions drawn by the authors and I have several concerns that should be addressed before this work could be considered acceptable for publication.

Major comments

1- The main behavioral characterization of this work is the enhanced response to restraint-induced mild stress (similar to the one described in the mdx mouse). Unfortunately, this test was only performed in 6-wk old animals. While the authors explain that they “used this age in order to assess if the response to stress is affected at an early timepoint”, it is also necessary to include later time points. Especially considering that other tests were performed in older animals, it is extremely surprising to not show this test in 3 month- and 7 month-old animals. These data should absolutely be included since 1) it is known that the response to mild stress increases with age in mdx mice and 2) if this model of rat is used for therapeutic purpose, it is very likely that the tests may be used at much later time points than 6 weeks.

2- The conclusion of the authors that the DMD rat model does not show major cognitive deficit is not supported by their experimental data. Learning and memory tasks need to be performed to document this assumption. Tests previously used in the mdx mouse, which holds comparable genetic defect, should be used for comparison. Moreover, the underlying mechanisms of the enhanced fear response to mild stress are not characterized, but studies in mdx mice suggest they may involve altered cognitive processes.

3- Anxiety: Using only the elevated plus-maze test is inadequate to conclude that the DMD rats do not exhibit enhanced anxiety. Indeed, although mdx mice show normal behaviour in plus maze, other studies indicate a deficit in the light-dark choice anxiety test (Remmelink et al., 2016; Chaussenot and Vaillend, 2017). As the DMD rat shows enhanced fear responses comparable with mdx mice, one may expect similar phenotypes in other tests not included in the present study. At least the light-dark test should be performed before concluding on anxiety in this model. Moreover, please indicate the intensity of the light which participates as an aversive stimulus to induce anxiety, and refer to publication using same conditions and demonstrating the protocol induces anxiety in rats.

4- The authors insist on the effect observed in the ledged beam-walking test but the experimental data do not support this conclusion that strongly, and the conclusions are therefore misleading. For example: the authors write: “this was correlated with a non-significant increase in the total number of steps to cross the beam (p = 0.1229, Fig. 1B)” and just after they conclude “This indicates that mutant animals spend more time to perform this task, with more and slower steps.” However, here there is no significant difference in the number of steps and on the figure the difference is minimal from a quantitative point of view (16 steps versus 17 steps). The conclusions should therefore be toned down.

5- Western blots and figure 5: there is, in addition to a normalization with the GAPDH, a normalization with a positive control named “Control” in fig5B, C and D. Hovere this is not the same control for Dp71, Dp140 and DP427 (as stated in the figre legend). For more clarity, it should be written on the figure what the positive control is (WT total brain for B and C, and WT skeletal muscle for D) and probably explained in the results section why this normalization is being carried out and what it provides.

Moreover, the authors affirm that there is no dp140 in the spinal cord, yet a small band is visible in the western blot, so this shows that there seems to be a slight expression and not an absence of expression of dp140 in the spinal cord.

6- Regarding motor deficit, the authors say: "Hence, motor deficit underlined here only seems to be a consequence of dystrophin absence in muscles from the limbs, and not in the brain”. This conclusion is false, indeed the cerebellum is involved in motor coordination and dystrophin is expressed in the cerebellum, so it is wrong to say that this is due only to muscular dystrophin. Moreover, the authors themselves show that the brain structure in which dp427 seems to be most expressed is the cerebellum. It is therefore inappropriate to conclude that cerebral dystrophin is not involved, especially since one of the tests used (the ledged beam-walking test) involves motor coordination and therefore potentially the cerebellum.

7- Introduction: The author should explain why the rat model could be more suitable than mouse models for preclinical studies

8- The number of tested animals appears different depending on the test (n=8 and 10 in figure 1, then n=7 and 6 in fig2, etc). N numbers should be clearly indicated in all figure legends.

Minor:

1- Introduction : « abnormal distribution of GABAA receptors has also been found in brain of Duchenne patients [19], thus facilitating NMDA receptor-dependent synaptic plasticity and also inducing an abnormally increased hippocampal LTP [20] » ; here the authors mix data from patients and from mdx mice to give a general conclusion on patients. This is awkward as no LTP not NMDA studies have been done in patients.

2- Discussion: “The highest levels found in prefrontal cortex compared to amygdala, although non-significant, still tells us a possible explanation for the highest stress sensitivity for Dmdmdx rats”; this is a highly speculative assumption. Also, “levels of other dystrophin isoforms Dp140, expressed in smooth muscle layer of blood vessels and Dp71, exclusively expressed in endothelial cells of blood capillaries from both groups”: It has not been demonstrated that Dp140 is expressed in smooth muscles nor that Dp71 is expressed in endothelial cells (rather, many publications indicate localization in glial endfeet contacting blood vessels).

3- Figure 3 and 4 were inversed in the manuscript

Reviewer #2: The manuscript describes a series of experiments designed to assess the behavioural phenotype of the Dmdmdx rat model of Duchenne muscular dystrophy (DMD). The results show a clear locomoter deficit but no clear behavioural defects, other than a marked freezing response following mild handling stress, similar to that described in the mdx mouse model of DMD. The experiments appear competently performed and sufficiently powered (in most cases), although the 7 month old Dmdmdx rats showed a great degree of variation in the sucrose and water consumption assays and consequently these specific studies were underpowered.

The data presented in this paper support assays for future studies of therapeutics targeting muscle and brain function in this rat model.

Specific questions for the authors:

1. What sex of rats was used for the experiments? If both sexes were used, please show the number of each sex in each experiment.

2. Please give details of the diet for the rats (as suggested by the ARRIVE guidelines).

3. Were the video analyses of behaviour analysed blind to animal identity?

6. PLOS authors have the option to publish the peer review history of their article (what does this mean?). If published, this will include your full peer review and any attached files.

Reviewer #1: No

Reviewer #2: No

---

## [Author Response · Author response to Decision Letter 0]

28 Jan 2020

We thank the reviewer for his positive feedback and his/her constructive comments. We understand his concerns and hope that the answers that we give below will satisfy him.

Major comments

1- The main behavioral characterization of this work is the enhanced response to restraint-induced mild stress (similar to the one described in the mdx mouse). Unfortunately, this test was only performed in 6-wk old animals. While the authors explain that they “used this age in order to assess if the response to stress is affected at an early timepoint”, it is also necessary to include later time points. Especially considering that other tests were performed in older animals, it is extremely surprising to not show this test in 3 month- and 7 month-old animals. These data should absolutely be included since 1) it is known that the response to mild stress increases with age in mdx mice and 2) if this model of rat is used for therapeutic purpose, it is very likely that the tests may be used at much later time points than 6 weeks.

Indeed, restraint-induced mild stress is only performed in 6 weeks young animals, as it was the objective of this study. The other tests included in this study, performed in older animals, were done on a previous cohort of animals, more than a year before thinking about performing restraint stress on these animals. This explains why we did not study response to stress in older animals. We deeply agree that this should be done in the future on other cohorts of rats (3 and 7 months). The objective of this work is to offer answers to people asking questions about the quantification and evaluation of brain dystrophin. Moreover, we found an early hyper responsivity to stress during the behavioral evaluation, so we thought it was important to quickly communicate about this finding to teams already using this model. This is a first step characterizing neuronal aspects on this model that was only previously studied for muscle and heart disease aspects, and we think that this is an important message to share, without waiting for data at more advanced ages, that will be long to obtain. We have added a few words about this aspect in the discussion, from lines 420 to 422: 

These analyses were performed on young animals at the age of 6 weeks, and we cannot exclude that this phenotype will be exacerbated at later timepoints, which has to be taken into account in the design and follow-up of preclinical studies using this model. In mdx mice, this behavior has been shown to increase with age and is thought to be underlined by an alteration of amygdala GABAergic circuits in dystrophin deficient animals [25].

2- The conclusion of the authors that the DMD rat model does not show major cognitive deficit is not supported by their experimental data. Learning and memory tasks need to be performed to document this assumption. Tests previously used in the mdx mouse, which holds comparable genetic defect, should be used for comparison. Moreover, the underlying mechanisms of the enhanced fear response to mild stress are not characterized, but studies in mdx mice suggest they may involve altered cognitive processes.

As previously explained, cognitive exploration was not extensively performed on these animals, but this is planned for a later study, especially in order to characterize fear response following stress. We have therefore removed mentions about cognition in the paper.

3- Anxiety: Using only the elevated plus-maze test is inadequate to conclude that the DMD rats do not exhibit enhanced anxiety. Indeed, although mdx mice show normal behaviour in plus maze, other studies indicate a deficit in the light-dark choice anxiety test (Remmelink et al., 2016; Chaussenot and Vaillend, 2017). As the DMD rat shows enhanced fear responses comparable with mdx mice, one may expect similar phenotypes in other tests not included in the present study. At least the light-dark test should be performed before concluding on anxiety in this model. Moreover, please indicate the intensity of the light which participates as an aversive stimulus to induce anxiety, and refer to publication using same conditions and demonstrating the protocol induces anxiety in rats.

We agree that EPM and dark-light box assess two different types of anxiety. EPM measures acrophobia-induced anxiety (fear of height) and light-dark test evaluates photophobia-induced anxiety (repulsion for light). Since rodents are naturally acrophobic and photophobic, these tests are good models to study anxiety in rats. As these tests focus on different kinds of anxiety, they can dissociate. One may find a pathological phenotype in one of the tests but not in the others.

In the light-dark box test and in the EPM, animals have a place where to hide, whereas in the open-field, the whole arena is exposed to light, so the open field may be considered as more stressful than the light-dark box of EPM, especially if the brightness is the same in the open field as in the light compartment of the light-dark box. In this study, for the open-field, light intensity was set at 100 Lux, as the main objective was not to study anxiety, but rather evaluate locomotion and response to restraint-stress. In the EPM, light intensity was higher (300 Lux). These details have been added in the manuscript on line 158 and 184.

4- The authors insist on the effect observed in the ledged beam-walking test but the experimental data do not support this conclusion that strongly, and the conclusions are therefore misleading. For example: the authors write: “this was correlated with a non-significant increase in the total number of steps to cross the beam (p = 0.1229, Fig. 1B)” and just after they conclude “This indicates that mutant animals spend more time to perform this task, with more and slower steps.” However, here there is no significant difference in the number of steps and on the figure the difference is minimal from a quantitative point of view (16 steps versus 17 steps). The conclusions should therefore be toned down.

We agree with this remark concerning a mitigated effect, therefore we have simply removed the last sentence in this result part, on lines 262 and 263.

5- Western blots and figure 5: there is, in addition to a normalization with the GAPDH, a normalization with a positive control named “Control” in fig5B, C and D. Hovere this is not the same control for Dp71, Dp140 and DP427 (as stated in the figure legend). For more clarity, it should be written on the figure what the positive control is (WT total brain for B and C, and WT skeletal muscle for D) and probably explained in the results section why this normalization is being carried out and what it provides.

Moreover, the authors affirm that there is no dp140 in the spinal cord, yet a small band is visible in the western blot, so this shows that there seems to be a slight expression and not an absence of expression of dp140 in the spinal cord.

We have modified Fig 5 by adding the type of control used for Dp71, Dp140 and Dp427 normalization: total muscle (Dp427) or total brain (Dp71 and Dp140) of WT animal. Also, in the Methods section (not the Results section as suggested by the reviewer), we have explained why we used either brain or muscle for this normalization, from lines 327 to 330:

A normalization step was added by using total WT rat brain for Dp71 and Dp140, because these isoforms are absent from WT muscles, and by using WT skeletal muscle extract for Dp427 quantification, this isoform being highly expressed in muscle.

Concerning Dp140 in spinal cord, this comment is true, so we have changed this point in the text on line 341.

6- Regarding motor deficit, the authors say: "Hence, motor deficit underlined here only seems to be a consequence of dystrophin absence in muscles from the limbs, and not in the brain”. This conclusion is false, indeed the cerebellum is involved in motor coordination and dystrophin is expressed in the cerebellum, so it is wrong to say that this is due only to muscular dystrophin. Moreover, the authors themselves show that the brain structure in which dp427 seems to be most expressed is the cerebellum. It is therefore inappropriate to conclude that cerebral dystrophin is not involved, especially since one of the tests used (the ledged beam-walking test) involves motor coordination and therefore potentially the cerebellum.

This is a true and very relevant comment. We have revised the text accordingly in the discussion part from lines 376 to 378:

Motor deficit underlined here may therefore be a consequence of dystrophin absence in muscles from the limbs, and / or of reduced levels of cerebellar dystrophin, as demonstrated by western blot. 

7- Introduction: The author should explain why the rat model could be more suitable than mouse models for preclinical studies

An explanation has been added on lines 109 to 115:

In this study, we used the Dmdmdx rat model, which was recently generated [26] in order to counteract the minor clinical dysfunction of mdx mouse [27] and the fact that their small size imposes limitations in the analysis of several aspects of the disease. Moreover, one of the advantages of rat over mice in preclinical studies is that rat behavior is much better characterized. They have finer and more accurate motor coordination than mice and exhibit a richer behavioral display, including more complex social traits. Moreover, rats have a convenient size since they are 10 times larger than mice but are still a small laboratory animal model and allow studies with high statistical power.

8- The number of tested animals appears different depending on the test (n=8 and 10 in figure 1, then n=7 and 6 in fig2, etc). N numbers should be clearly indicated in all figure legends.

We have indeed added the N values in each figure legend.

Minor:

1- Introduction : « abnormal distribution of GABAA receptors has also been found in brain of Duchenne patients [19], thus facilitating NMDA receptor-dependent synaptic plasticity and also inducing an abnormally increased hippocampal LTP [20] » ; here the authors mix data from patients and from mdx mice to give a general conclusion on patients. This is awkward as no LTP not NMDA studies have been done in patients.

We totally agree with the reviewer, as it is a mistake from our side that slipped through our internal correction process. We have therefore modified this part from lines 92 to 94: 

[…] like an abnormal synaptic clustering and density of GABAA receptors in CA1 hippocampal dendritic layer [13, 16-18], thus facilitating NMDA receptor-dependent synaptic plasticity and also inducing an abnormally increased hippocampal LTP [19]. We have to note, as an aside, that t the clinical level, an abnormal distribution of GABAA receptors has also been found in brain of Duchenne patients [20] […]

2- Discussion: “The highest levels found in prefrontal cortex compared to amygdala, although non-significant, still tells us a possible explanation for the highest stress sensitivity for Dmdmdx rats”; this is a highly speculative assumption. Also, “levels of other dystrophin isoforms Dp140, expressed in smooth muscle layer of blood vessels and Dp71, exclusively expressed in endothelial cells of blood capillaries from both groups”: It has not been demonstrated that Dp140 is expressed in smooth muscles nor that Dp71 is expressed in endothelial cells (rather, many publications indicate localization in glial endfeet contacting blood vessels).

We have toned down the first speculation concerning the comparison between prefrontal cortex and amygdala from lines 448 to 452:

Moreover, although quite speculative at the moment, the highest levels found in prefrontal cortex compared to amygdala, could tell us a possible explanation for the highest stress sensitivity for Dmdmdx rats. Indeed, as prefrontal cortex and amygdala are known to communicate and be both involved in stress response [49, 50],

Concerning the second comment, it is strictly based on our histology observations (Fig. 6), where the immunolabelling was very clear on the endothelium and the smooth muscles (including for the vessels outside the nervous tissue like the meninges, therefore showing that it is not associated with glial cells, at least for this localization). For the smallest vessels in the nervous tissue, glial cells and endothelial cells are very close, so it has been interpreted as endothelial tissue but we maybe cannot exclude the presence of a glial marking around the vessels. This remark has been added on line 355.

3- Figure 3 and 4 were inversed in the manuscript

Actually, the name of the figures submitted during the submission process was reversed. The nomenclature in the paper is indeed the correct one.

Reviewer #2: The manuscript describes a series of experiments designed to assess the behavioural phenotype of the Dmdmdx rat model of Duchenne muscular dystrophy (DMD). The results show a clear locomoter deficit but no clear behavioural defects, other than a marked freezing response following mild handling stress, similar to that described in the mdx mouse model of DMD. The experiments appear competently performed and sufficiently powered (in most cases), although the 7 month old Dmdmdx rats showed a great degree of variation in the sucrose and water consumption assays and consequently these specific studies were underpowered.

We agree with this point, a high variability is observed, as already mentioned in the text.

The data presented in this paper support assays for future studies of therapeutics targeting muscle and brain function in this rat model.

We thank the reviewer for this positive feedback.

Specific questions for the authors:

1. What sex of rats was used for the experiments? If both sexes were used, please show the number of each sex in each experiment.

All animals used were males. This has been added in the text on line 137:

The rats were housed in a controlled environment (ventilated racks, ambient temperature of 21°C, ambient hygrometry of 55%, 12 h light/dark cycle (dark at 8 pm, light at 8 am)), with several animals per cage, all males.

2. Please give details of the diet for the rats (as suggested by the ARRIVE guidelines).

Thanks for pointing us to these guidelines, that are indeed really useful when reporting work performed with animals. Details on the diet have been added in the Methods section from line 137 to 139:

Diet consisted of a standard diet (SAFE A04, Safe, Augy, France) given ad libitum, sterilized and filtrated water also given ad libitum.

3. Were the video analyses of behaviour analysed blind to animal identity?

Yes, all analyses were performed blind the animal identities. This has been added to the text on line 139:

All behavioral tests were performed blind to animal identities.

---

## [Decision Letter · Decision Letter 1]

10 Feb 2020

PONE-D-19-28211R1

Dystrophin-deficient Dmdmdx rat model displays an increased behavioral response to restraint-induced mild stress

PLOS ONE

Dear Dr. Caudal,

Thank you for submitting your revised manuscript to PLOS ONE. The revision has been evaluated again by the reviewer who originally suggested a major revision. As you can see, your responses have satisfactory addressed the major concerns that had been raised. However, a few suggestions for further improvement came up, which seem feasible. Therefore, we invite you to submit a re-revised version of the manuscript that addresses these additional points .

To enhance the reproducibility of your results, we recommend that if applicable you deposit your laboratory protocols in protocols.io, where a protocol can be assigned its own identifier (DOI) such that it can be cited independently in the future. For instructions see: http://journals.plos.org/plosone/s/submission-guidelines#loc-laboratory-protocols

We look forward to receiving your revised manuscript.

Kind regards,

Gerhard Wiche, Ph.D.

Academic Editor

PLOS ONE

Reviewers' comments:

Reviewer's Responses to Questions

**Comments to the Author**

1. If the authors have adequately addressed your comments raised in a previous round of review and you feel that this manuscript is now acceptable for publication, you may indicate that here to bypass the “Comments to the Author” section, enter your conflict of interest statement in the “Confidential to Editor” section, and submit your "Accept" recommendation.

Reviewer #1: (No Response)

2. Is the manuscript technically sound, and do the data support the conclusions?

Reviewer #1: Partly

3. Has the statistical analysis been performed appropriately and rigorously? 

Reviewer #1: Yes

4. Have the authors made all data underlying the findings in their manuscript fully available?

Reviewer #1: Yes

5. Is the manuscript presented in an intelligible fashion and written in standard English?

Reviewer #1: Yes

6. Review Comments to the Author

Reviewer #1: (No Response)

7. PLOS authors have the option to publish the peer review history of their article (what does this mean?). If published, this will include your full peer review and any attached files.

Reviewer #1: No

---

## [Author Response · Author response to Decision Letter 1]

19 Feb 2020

PONE-D-19-28211

Dystrophin-deficient Dmdmdx rat model displays an increased behavioral response to

restraint-induced mild stress

Reviewer #1: This work by Caudal et al focuses on the behavioral characterization of the

dystrophin deficient DMD rat model previously described by some of the authors.

The manuscript is relatively well written, with experiments and results well designed and easy

to follow. The figures are also very clear. Overall, this is an interesting study which confirms

a behavioral phenotype observed in another animal model of DMD, the mdx mouse and is

therefore of interest to the research community in the DMD field.

However, the authors draw conclusions that may be premature based on the presented results.

Some experiments do not strongly support the conclusions drawn by the authors and I have

several concerns that should be addressed before this work could be considered acceptable for

publication.

We thank the reviewer for his positive feedback and his/her constructive comments. We

understand his concerns and hope that the answers that we give below will satisfy him.

Major comments

1- The main behavioral characterization of this work is the enhanced response to restraint induced mild stress (similar to the one described in the mdx mouse). Unfortunately, this test

was only performed in 6-wk old animals. While the authors explain that they “used this age in

order to assess if the response to stress is affected at an early timepoint”, it is also necessary to

include later time points. Especially considering that other tests were performed in older

animals, it is extremely surprising to not show this test in 3 month- and 7 month-old animals.

These data should absolutely be included since 1) it is known that the response to mild stress

increases with age in mdx mice and 2) if this model of rat is used for therapeutic purpose, it is

very likely that the tests may be used at much later time points than 6 weeks.

Indeed, restraint-induced mild stress is only performed in 6 weeks young animals, as it was

the objective of this study. The other tests included in this study, performed in older animals,

were done on a previous cohort of animals, more than a year before thinking about

performing restraint stress on these animals. This explains why we did not study response to

stress in older animals. We deeply agree that this should be done in the future on other cohorts

of rats (3 and 7 months). The objective of this work is to offer answers to people asking

questions about the quantification and evaluation of brain dystrophin. Moreover, we found an

early hyper responsivity to stress during the behavioral evaluation, so we thought it was

important to quickly communicate about this finding to teams already using this model. This

is a first step characterizing neuronal aspects on this model that was only previously studied

for muscle and heart disease aspects, and we think that this is an important message to share,

without waiting for data at more advanced ages, that will be long to obtain. We have added a

few words about this aspect in the discussion, from lines 420 to 422:

The reviewer finds this answer acceptable however since 1) the authors confirm that “The objective of this work is to offer answers about the quantification and evaluation of brain dystrophin” and 2) agree that the restraint-induced mild stress should be done in older animals to fully characterize this phenotype, the title of the manuscript should be changed to a more general title (not only focusing on the behavioral response to restraint-induced mild stress which is only partially characterized here).

Maybe something like: “Characterization of brain dystrophins absence and its impact in Dystrophin-deficient Dmdmdx rat model” or

“A first characterization of the impact of brain Dp47 dystrophin deficiency in the dystrophin-deficient Dmdmdx rat model”. (these are just suggestions)

We agree with the reviewer and have therefore modified the title accordingly.

2- The conclusion of the authors that the DMD rat model does not show major cognitive

deficit is not supported by their experimental data. Learning and memory tasks need to be

performed to document this assumption. Tests previously used in the mdx mouse, which

holds comparable genetic defect, should be used for comparison. Moreover, the underlying

mechanisms of the enhanced fear response to mild stress are not characterized, but studies in

mdx mice suggest they may involve altered cognitive processes.

As previously explained, cognitive exploration was not extensively performed on these

animals, but this is planned for a later study, especially in order to characterize fear response

following stress. We have therefore removed mentions about cognition in the paper.

The authors took into consideration the criticisms concerning the cognitive part of the disease, they decided to nuance their remarks. OK.

3- Anxiety: Using only the elevated plus-maze test is inadequate to conclude that the DMD

rats do not exhibit enhanced anxiety. Indeed, although mdx mice show normal behaviour in

plus maze, other studies indicate a deficit in the light-dark choice anxiety test (Remmelink et

al., 2016; Chaussenot and Vaillend, 2017). As the DMD rat shows enhanced fear responses

comparable with mdx mice, one may expect similar phenotypes in other tests not included in

the present study. At least the light-dark test should be performed before concluding on

anxiety in this model. Moreover, please indicate the intensity of the light which participates as

an aversive stimulus to induce anxiety, and refer to publication using same conditions and

demonstrating the protocol induces anxiety in rats.

We agree that EPM and dark-light box assess two different types of anxiety. EPM measures

acrophobia-induced anxiety (fear of height) and light-dark test evaluates photophobia-induced

anxiety (repulsion for light). Since rodents are naturally acrophobic and photophobic, these

tests are good models to study anxiety in rats. As these tests focus on different kinds of

anxiety, they can dissociate. One may find a pathological phenotype in one of the tests but not

in the others.

In the light-dark box test and in the EPM, animals have a place where to hide, whereas in the

open-field, the whole arena is exposed to light, so the open field may be considered as more

stressful than the light-dark box of EPM, especially if the brightness is the same in the

open field as in the light compartment of the light-dark box. In this study, for the open-field,

light intensity was set at 100 Lux, as the main objective was not to study anxiety, but rather

evaluate locomotion and response to restraint-stress. In the EPM, light intensity was higher

(300 Lux). These details have been added in the manuscript on line 158 and 184.

The reviewer accepts the authors answer but a comment about previous studies in mdx mice should be added in the discussion where they mention:

 “Contrasting results have been found on anxiety in the classical model of mdx mice. Indeed, in 2009 a group showed that this model did not display an anxiety-like phenotype in the elevated plus maze (24). However, other recent studies indicate a deficit in the light-dark choice anxiety test (Remmelink et al., 2016; Chaussenot and Vaillend, 2017). Moreover, another group demonstrated anxiety-like and depression-like behaviors in mdx mice, associated with decreased BDNF (Brain derived neurotrophic factor) levels [42]. It may therefore be useful to evaluate BDNF levels and perform light-dark choice anxiety tests to fully characterise the anxiety phenotype in Dmdmdx rats.” 

Please amend accordingly.

We have amended the text according to reviewer’s comment.

4- The authors insist on the effect observed in the ledged beam-walking test but the

experimental data do not support this conclusion that strongly, and the conclusions are

therefore misleading. For example: the authors write: “this was correlated with a non significant increase in the total number of steps to cross the beam (p = 0.1229, Fig. 1B)” and just after they conclude “This indicates that mutant animals spend more time to perform this

task, with more and slower steps.” However, here there is no significant difference in the

number of steps and on the figure the difference is minimal from a quantitative point of view

(16 steps versus 17 steps). The conclusions should therefore be toned down.

We agree with this remark concerning a mitigated effect, therefore we have simply removed

the last sentence in this result part, on lines 262 and 263.

Ok, but please also remove the following sentence from the discussion: “The present results demonstrate that Dmdmdx rats display neuromotor alterations at 7 months, as shown with the transversal beam test, in which mutant animals spent more time to cross the beam, as they did slightly more steps to reach the other extremity.

We have amended the text according to reviewer’s comment.

5- Western blots and figure 5: there is, in addition to a normalization with the GAPDH, a

normalization with a positive control named “Control” in fig5B, C and D. However this is not

the same control for Dp71, Dp140 and DP427 (as stated in the figure legend). For more

clarity, it should be written on the figure what the positive control is (WT total brain for B and

C, and WT skeletal muscle for D) and probably explained in the results section why this

normalization is being carried out and what it provides.

Moreover, the authors affirm that there is no dp140 in the spinal cord, yet a small band is

visible in the western blot, so this shows that there seems to be a slight expression and not an

absence of expression of dp140 in the spinal cord.

We have modified Fig 5 by adding the type of control used for Dp71, Dp140 and Dp427

normalization: total muscle (Dp427) or total brain (Dp71 and Dp140) of WT animal. Also, in the Methods section (not the Results section as suggested by the reviewer), we have explained why we used either brain or muscle for this normalization, from lines 327 to 330:

A normalization step was added by using total WT rat brain for Dp71 and Dp140, because

these isoforms are absent from WT muscles, and by using WT skeletal muscle extract for Dp427 quantification, this isoform being highly expressed in muscle.

Concerning Dp140 in spinal cord, this comment is true, so we have changed this point in the

text on line 341.

Reviewer accepts this change, however still finds unfortunate that the authors do not have the same control tissues for the different tissues analyzed. Concerning cerebral Dp427, why did they use a WT control from muscle tissue since Dp427 is expressed in the CNS? This is what the reviewer implied when asking “explain in the results section why this normalization is being carried out and what it provides”. Did the author intend to comment on the difference of expression between muscle and brain? Because if no explanation is given, it seems inappropriate to use muscle as control (since DP427 is expressed in total brain).

This muscle control was used as an internal control for Dp427. Indeed, at the beginning of western blot experiments, we had no idea how Dp427 would appear on the gels, therefore we had to choose an internal control that was used routinely in our lab. This allowed us to validate our brain dystrophin extraction protocol, protein dosing, and also, and more importantly revelation by ECL. Indeed, as we did several gels, and therefore several films, the exposure time could vary from a few seconds betwwen films, so this well established internal control was important for us. The objective here is not to compare Dp427 levels between brain and muscles, but rather to remove an experimental bias that could occur before we knew anything about rat brain dystrophin western blotting study in our samples with our experimental conditions. Moreover, we have cautioulsy performed another normalization with GAPDH. 

6- Regarding motor deficit, the authors say: "Hence, motor deficit underlined here only seems

to be a consequence of dystrophin absence in muscles from the limbs, and not in the brain”.

This conclusion is false, indeed the cerebellum is involved in motor coordination and

dystrophin is expressed in the cerebellum, so it is wrong to say that this is due only to

muscular dystrophin. Moreover, the authors themselves show that the brain structure in which

dp427 seems to be most expressed is the cerebellum. It is therefore inappropriate to conclude

that cerebral dystrophin is not involved, especially since one of the tests used (the ledged

beam-walking test) involves motor coordination and therefore potentially the cerebellum.

This is a true and very relevant comment. We have revised the text accordingly in the

discussion part from lines 376 to 378:

Motor deficit underlined here may therefore be a consequence of dystrophin absence in muscles from the limbs, and / or of reduced levels of cerebellar dystrophin, as demonstrated by western blot.

The authors have taken into consideration the remarks concerning the possible impact of cerebellum in the motor dysfonction and have revised the text. The reviewer suggests to rephrase as follow for more clarity: 

“Motor deficit underlined here may be a consequence of dystrophin absence in muscles

from the limbs, but we cannot exclude they may also result from reduced levels of cerebellar dystrophin, as demonstrated by western blot.”

We have amended the text according to reviewer’s comment.

7- Introduction: The author should explain why the rat model could be more suitable than

mouse models for preclinical studies

An explanation has been added on lines 109 to 115:

In this study, we used the Dmdmdx rat model, which was recently generated [26] in order to

counteract the minor clinical dysfunction of mdx mouse [27] and the fact that their small size

imposes limitations in the analysis of several aspects of the disease. Moreover, one of the

advantages of rat over mice in preclinical studies is that rat behavior is much better

characterized. They have finer and more accurate motor coordination than mice and exhibit a

richer behavioral display, including more complex social traits. Moreover, rats have a

convenient size since they are 10 times larger than mice but are still a small laboratory animal model and allow studies with high statistical power.

Ok but Authors should supplement their arguments with bibliographical references. Specifically to argue that rat behavior is better characterized and has richer behavioral display, which may not be obvious regarding the major advances made in characterizing complex cognitive functions in transgenic mice since the 90’s. In addition, it would be appreciated if they could better detail the positive points of their model compared respectively to the murine model (how is being 10 times larger better?)

We have removed the sentence concerning rat behavior which is indeed a bit exaggerated. The nex text is now: 

In this study, we used the Dmdmdx rat model, which was recently generated [26] in order to counteract the minor clinical dysfunction of mdx mouse [27] and the fact that their small size imposes limitations in the analysis of several aspects of the disease. Moreover, rats display complex social traits and have a convenient size since they are 10 times larger than mice, allowing the possibility to collect large quantities of biological tissues compared to mice. But rats remain a small laboratory animal model and allow studies with high statistical power.

8- The number of tested animals appears different depending on the test (n=8 and 10 in figure

1, then n=7 and 6 in fig2, etc). N numbers should be clearly indicated in all figure legends.

We have indeed added the N values in each figure legend.

OK

Minor:

1- Introduction : « abnormal distribution of GABAA receptors has also been found in brain of

Duchenne patients [19], thus facilitating NMDA receptor-dependent synaptic plasticity and

also inducing an abnormally increased hippocampal LTP [20] » ; here the authors mix data

from patients and from mdx mice to give a general conclusion on patients. This is awkward as

no LTP not NMDA studies have been done in patients.

We totally agree with the reviewer, as it is a mistake from our side that slipped through our

internal correction process. We have therefore modified this part from lines 92 to 94:

[…] like an abnormal synaptic clustering and density of GABAA receptors in CA1 hippocampal dendritic layer [13, 16-18], thus facilitating NMDA receptor-dependent synaptic plasticity and also inducing an abnormally increased hippocampal LTP [19]. We have to note, as an aside, that t the clinical level, an abnormal distribution of GABAA receptors has also been found in brain of Duchenne patients [20] […]

OK

2- Discussion: “The highest levels found in prefrontal cortex compared to amygdala, although

non-significant, still tells us a possible explanation for the highest stress sensitivity for

Dmdmdx rats”; this is a highly speculative assumption. Also, “levels of other dystrophin

isoforms Dp140, expressed in smooth muscle layer of blood vessels and Dp71, exclusively

expressed in endothelial cells of blood capillaries from both groups”: It has not been

demonstrated that Dp140 is expressed in smooth muscles nor that Dp71 is expressed in

endothelial cells (rather, many publications indicate localization in glial endfeet contacting

blood vessels).

We have toned down the first speculation concerning the comparison between prefrontal

cortex and amygdala from lines 448 to 452:

Moreover, although quite speculative at the moment, the highest levels found in prefrontal

cortex compared to amygdala, could tell us a possible explanation for the highest stress

sensitivity for Dmdmdx rats. Indeed, as prefrontal cortex and amygdala are known to

communicate and be both involved in stress response [49, 50],

Reviewer still feels that this is highly speculative and based on a non-significant result, and therefore advises to remove the entire sentence.

As suggested, we have removed the speculative part of the sentence, but not the part mentioning the relationship between both regions, which is still true:

In WT animals, the highest levels of Dp427 isoforms were detected in the cerebellum and spinal cord, which is in agreement with studies demonstrating strong dystrophin expression in cerebellum, but also cortex and hippocampus in WT mice and rats [48, 49]. As prefrontal cortex and amygdala are known to communicate and be both involved in stress response [50, 51], dystrophin levels in these brain areas might participate in a typical adaptive stress response for WT rats.

Concerning the second comment, it is strictly based on our histology observations (Fig. 6),

where the immunolabelling was very clear on the endothelium and the smooth muscles

(including for the vessels outside the nervous tissue like the meninges, therefore showing that

it is not associated with glial cells, at least for this localization). For the smallest vessels in the

nervous tissue, glial cells and endothelial cells are very close, so it has been interpreted as

endothelial tissue but we maybe cannot exclude the presence of a glial marking around the

vessels. This remark has been added on line 355.

The reviewer acknowledges that the authors find that the “immunolabelling was very clear on the endothelium and the smooth muscles (including for the vessels outside the nervous tissue like the meninges, therefore showing that it is not associated with glial cells, at least for this localization)”, however it cannot be stated without showing either the data mentioned by the authors, or co-staining data of endothelium and smooth muscles to conclude about the expression of dystrophin there. 

As mentioned by the authors themselves, the main objective of this work is to characterize the expression and localization of dystrophin in the brain of this novel dmd-mdx rat model and the reviewer completely agrees that this work is very important and will likely serve as reference for future work in the field. As such, it is crucial to characterize properly the localization of the various isoforms using appropriate controls.

Therefore, in addition to adding co-staining, it is also necessary to better explain the choice of antibodies as this may be a little confusing for the reader:

-Dys2 was chosen to detect Dp71 but as a C-ter Ab, it will stain all isoforms of dystrophin (including Dp427 and Dp140 – even in the dmd-mdx model for the latter one). The authors should therefore provide controls that the staining shown for Dp71 is not Dp427 or Dp140 staining (for Dp427, this can be compared with dmd-mdx tissues and explained in the text but for Dp140, how can the authors be sure that this is not Dp140 staining for example?)

-Similar point for Mannex5556 which is specific of exon55-56 and which should stain both Dp140 and Dp427. The authors should better explain and justify why they consider the staining to be Dp140 specific. 

Also, since it is the first time that Manex5556 is used in rats (at least to the reviewer’s knowledge), it would be better to provide at least one negative control without primary antibody as supplemental data.

Maybe this was not clear in the text, but the main objective of this immunolabelling with these 3 antibodies is of course not to be specific of a given isoform, we agree with the reviewer on this aspect, despite immunolabelling were different between three antibodies and between two genotypes, indicating that they recognize different epitopes. These antibodies have been chosen to cover three different regions of dystrophin protein, not to be specific of a given region. As such, we have modified the text in the Methods section accordingly to erase the idea that these antibodies are specific. Concerning co-staining, it is rather difficult to perform with our monoclonal antibodies, and we have to mention that all these analyses and distinction between endothelial cells and smooth muscles is very typical and was performed by a skilled anatomopatholgist. We have added this sentence at the end of Methods section:“All slides evaluations were performed by a skilled pathologist certified by the European College of Veterinary Pathology.”

We also add below, as requested by the reviewer, a table showing the negative controls (x20) that were performed for the 3 antibodies in parallel, but we don’t think this needs to be added into a supplemental figure. Light non-specific binding is detected on some collagen fibers. 

 Negative control for NCL-DYSB

 Negative control for MANEX5556

 Negative control for NCL-DYS2

The addition of a schematic representation of the dystrophin epitopes recognized by the different antibodies used would be very useful here. 

We have added a figure in order to show this representation (Figure 7).

3- Figure 3 and 4 were inversed in the manuscript

Actually, the name of the figures submitted during the submission process was reversed. The

nomenclature in the paper is indeed the correct one.

OK

---

## [Editor Report · Decision Letter 2]

21 Feb 2020

Characterization of brain dystrophins absence and impact in dystrophin-deficient Dmdmdx rat model

PONE-D-19-28211R2

Dear Dr. Caudal,

We are pleased to inform you that your manuscript has been judged scientifically suitable for publication and will be formally accepted for publication once it complies with all outstanding technical requirements.

With kind regards,

Gerhard Wiche, Ph.D.

Academic Editor

PLOS ONE
---

## [Editor Report · Acceptance letter]

24 Feb 2020

PONE-D-19-28211R2 

Characterization of brain dystrophins absence and impact in dystrophin-deficient Dmdmdx rat model 

Dear Dr. Caudal:

I am pleased to inform you that your manuscript has been deemed suitable for publication in PLOS ONE. Congratulations! Your manuscript is now with our production department. 

With kind regards,

on behalf of

Prof. Gerhard Wiche 

Academic Editor

PLOS ONE